# Proteomics Profiling of Bladder Cancer Tissues from Early to Advanced Stages Reveals NNMT and GALK1 as Biomarkers for Early Detection and Prognosis of BCa

**DOI:** 10.3390/ijms241914938

**Published:** 2023-10-06

**Authors:** Katarina Davalieva, Sanja Kiprijanovska, Ognen Ivanovski, Aleksandar Trifunovski, Skender Saidi, Aleksandar Dimovski, Zivko Popov

**Affiliations:** 1Research Centre for Genetic Engineering and Biotechnology “Georgi D Efremov”, Macedonian Academy of Sciences and Arts, 1000 Skopje, North Macedonia; skiprijanovska@manu.edu.mk (S.K.); a.dimovski@manu.edu.mk (A.D.); 2Clinical Centre “Mother Theresa”, University Clinic for Urology, 1000 Skopje, North Macedonia; ognen.ivanovski@medf.ukim.edu.mk (O.I.); trifunovskia@gmail.com (A.T.); skendersaidi@yahoo.com (S.S.); 3Faculty of Pharmacy, University “St. Cyril and Methodius”, 1000 Skopje, North Macedonia; 4Clinical Hospital “Acibadem Sistina”, 1000 Skopje, North Macedonia; zivkopopov2000@yahoo.com; 5Medical Faculty, University “St. Cyril and Methodius”, 1000 Skopje, North Macedonia; 6Macedonian Academy of Sciences and Arts, 1000 Skopje, North Macedonia

**Keywords:** bladder cancer, tissue, urine, biomarker, LC-MS/MS, proteomics, ELISA

## Abstract

The high recurrence rate and invasive diagnostic and monitoring methods in bladder cancer (BCa) clinical management require the development of new non-invasive molecular tools for early detection, particularly for low-grade and low-stage BCa as well as for risk stratification. By using an in-solution digestion method and label-free data-independent LC-MS/MS coupled with ion mobility, we profiled the BCa tissues from initiation to advanced stages and confidently identified and quantified 1619 proteins (≥2 peptides). A statistically significant difference in abundance (Anova ≤ 0.05) showed 494 proteins. Significant correlation with stage with steady up or down with BCa stages showed 15 proteins. Testing of NNMT, GALK1, and HTRA1 in urine samples showed excellent diagnostic potential for NNMT and GALK1 with AUC of 1.000 (95% CI: 1.000–1.000; *p* < 0.0001) and 0.801 (95% CI: 0.655–0.947; *p* < 0.0001), respectively. NNMT and GALK1 also showed very good potential in discriminating non-invasive low-grade from invasive high-grade BCa with AUC of 0.763 (95% CI: 0.606–0.921; *p* = 0.001) and 0.801 (95% CI: 0.653–0.950; *p* < 0.0001), respectively. The combination of NNMT and GALK1 increased prognostic accuracy (AUC = 0.813). Our results broaden the range of potential novel candidates for non-invasive BCa diagnosis and prognosis.

## 1. Introduction

Bladder cancer (BCa) is among the top 12 types of cancer detected worldwide, with more than 570,000 new cases annually and accounts for around 3% of all new cancer diagnoses and 2.1% of all cancer-associated deaths [1]. Approximately 90–95% of BCa tumors are urothelial cell carcinomas originating in the epithelium. Around 75% of newly diagnosed urothelial cell carcinomas are non-muscle invasive bladder cancer (NMIBC) characterized with stages pTa–pT1, while the remaining 25% are muscle-invasive bladder cancer (MIBC) characterized with stages ≥pT2 [2]. Patients with MIBC have poor prognosis with 5-year survival from 63% for T2 tumors to 46% for T3–T4 tumors and 15% for metastatic cancer [3]. Patients with NMIBC have high survival rates (Ta: 98%; T1: 88%) but also a high probability of disease recurrence (31–78%) and progression (up to 45%) within five years of disease diagnosis, with the highest rate observed in patients with high-grade T1 tumors (pT1G3) [4,5].

The current widely used diagnostic procedures for detecting BCa include urine cytology and cystoscopy, followed by biopsy if abnormal tissue is discovered. Urine cytology is inexpensive, non-invasive, and commonly used for initial BCa screening. However, the sensitivity of detecting low-grade tumors is poor, ranging from 10 to 43.6% [6]. Cystoscopy is the current ‘gold standard’ diagnostic procedure for BCa. However, it is associated with some serious drawbacks, starting from being invasive, expensive, and uncomfortable for the patient to missing up to 20% of papillary tumors (Ta and T1) and carcinoma in situ (CIS) [7,8,9]. To improve the rate and accuracy of detection, several new endoscopic technologies were developed, but these are invasive, expensive, time-consuming, and therefore, not able to significantly improve the diagnostic workflow [10]. As a result, the investigation for novel non-invasive biomarkers to detect BCa at early stages and reduce the need for surveillance cystoscopies and other invasive procedures increased over the last few decades [11,12,13,14]. As a result, several FDA-approved tests based on novel biomarkers have been introduced in the last years [13], but unfortunately this did not solve the problem of identifying low-grade tumors. Nevertheless, extensive research in this area, analyzing urine, tissue, blood, and extracellular vesicles has led to the identification of a great number of potential BCa biomarkers (extensively discussed in [11,12,13,14,15]). Biomarkers discovered in urine could have superior diagnostic sensitivity but often lack appropriate specificity. These biomarkers are often blood-associated proteins that are present in urine with altered abundancies due to the bleeding and angiogenesis mechanisms associated with BCa. The application of these biomarkers as BCa biomarkers is most likely limited to only high-risk patients and for monitoring of cancer recurrence. Another shortcoming of urine and other body fluids as a medium for the identification of high-sensitivity biomarkers is that proteins that are abundant in body fluids may mask, and thus limit the detection, of tissue-leakage proteins present at low concentrations.

On the other hand, exploring the tissue proteome of clinical tissue specimens is a straightforward strategy for discovering reliable BCa biomarker candidates. Tissue proteomics provides insight into the molecular mechanisms of BCa initiation and progression and, in addition to diagnostic biomarkers, could enable the identification of prognostic and predictive biomarkers. However, due to the invasive procedures for obtaining tissue as material for proteomics analysis and its limited availability, only a small number of studies investigated alterations in the tissue proteome of BCa patients. These studies yielded hundreds of differentially expressed proteins as potential biomarker candidates for studying the mechanism of BCa development and progression [12,15]. However, to confirm these findings and identify BCa-related biomarkers with high sensitivity and specificity, more well-designed tissue proteomics studies encompassing BCa tumors of all stages are needed.

In this study, we searched for potential bladder cancer biomarkers by profiling proteomic changes in tissue specimens of BCa ranging from Ta –T3 tumors using a strategy combining in-solution digestion method with two detergents (SDS and RapiGest) for sample preparation and label-free data-independent LC-MS/MS acquisition coupled with ion mobility. The generated quantitative tissue proteome was correlated with clinicopathological characteristics of the tumors that resulted in potential candidates for BCa progression. To assess the biomarker performance in terms of their sensitivity and specificity to detect BCa, as well as to give the prognosis in regard to the progression of the disease towards the invasive forms, three top candidates were tested in two sets of urine samples.

## 2. Results

### 2.1. Protein Identification

A total of 1847 proteins with quantitative values based on 25,381 peptides was identified by Progenesis QIP. The number of proteins and peptides was similar among the three compared groups (Mann–Whitney U-test, *p* > 0.05) (Figure 1A). We have identified 1277, 1285, and 1353 proteins in groups 1, 2 and 3, respectively, with 876 proteins common for all groups (Figure 1B). The correlation of normalized protein abundance across individual samples was very high with a median Spearman Rho correlation coefficient of 0.926 (Figure 1C). The hierarchical clustering of the proteins with normalized protein abundance showed a clear distinction between patients with non-invasive low-grade papillary urothelial carcinoma versus patients with invasive/infiltrative high-grade papillary urothelial carcinoma (Figure 1D).

### 2.2. Proteins with Differential Abundance among Groups

The protein identifications were further filtered to remove reverse sequences (*n* = 68), proteins identified on only one peptide (*n* = 159), and yeast ADH. The final report contained 1619 proteins identified based on ≥2 peptides (Appendix A). A statistically significant difference in abundance (Anova ≤ 0.05) showed 494 proteins (Appendix A). Significant differential abundance (Mann–Whitney U-test, *p* ≤ 0.05) in Group 1 vs. Group 2, Group 2 vs. Group 3, and Group 1 vs. Group 3 showed 340, 95, and 482 proteins, respectively (Table 1). After adjusting the *p*-values for multiple testing using the Benjamini-Hochberg (B-H) procedure and filtering the dataset for B-H *p* ≤ 0.05, significant differential abundance (Mann–Whitney U-test, *p* ≤ 0.05) showed 216 proteins (Group 1 vs. Group 2 = 81; Group 2 vs. Group 3 = 0; Group 1 vs. Group 3 = 195). The higher proportion of the differentially abundant proteins were down-regulated.

### 2.3. Functional Analysis

To gain insight into the cell/tissue origin and biological implication of the 494 proteins with significantly altered abundance among groups, we have analyzed the reported molecular functions, biological processes, cellular localization, the protein localization in the bladder and cancer in general (Appendix A), and performed an enrichment analysis with STRING. Bladder and cancer specificity were analyzed based on transcript detection according to the data available from the Human Protein Atlas (HPA).

All of the proteins for which there was available data in HPA had expression in the normal bladder tissue (434 proteins or 87.9%) (Figure 2A). Elevated expression in the bladder compared to other tissues had 7 proteins (1.4%), namely, OAS1, HPGD, S100P, TNFAIP2, KRT17, DHRS2 and PADI3. The remaining had elevated expression in other tissues but expressed in the bladder (170 proteins or 34.4%) and low tissue specificity but expressed in the bladder (257 proteins or 52%). Elevated expression in cancer had 56 proteins (11.3%) while 374 (75.7%) had low cancer specificity. From proteins elevated in cancer, 17 had elevated expression in urothelial cancer of which, 9 (16.1%) were elevated only in urothelial cancer (KRT7, GSTM1, HPGD, S100P, DHRS2, GDPD3, KRT23, TINAGL1, PADI3), while 8 (14.3%) in addition to urothelial cancer, were elevated in some other cancer too (SNCG, KRT13, KRT20, HSD17B2, FABP5, LY6D, INA, ANXA10).

The top-represented molecular functions were binding (GO:0005488), catalytic activity (GO:0003824), and structural molecule activity (GO:0005198) while the top-represented biological functions were cellular (GO:0009987) and metabolic processes (GO:0008152), biological regulation (GO:0065007) and localization (GO:0051179) (Figure 2B). The top represented protein classes were metabolite interconversion enzyme (PC00262) with more than 100 proteins, followed by cytoskeletal protein (PC00085) and protein modifying enzyme (PC00260). Among the top 5 associated pathways were the Integrin signaling pathway (P00034), Inflammation mediated by chemokine and cytokine signaling pathway (P00031), Cytoskeletal regulation by Rho GTPase (P00016), Wnt signaling pathway (P00057), and Glycolysis (P00024).

Protein-protein interaction analysis with STRING indicated that these proteins have more interactions among themselves than what would be expected for a random set of proteins of the same size and degree distribution drawn from the genome (PPI enrichment *p*-value = 1.0 × 10^−16^). Such an enrichment indicates that the proteins are at least partially biologically connected as a group (Figure 2C). There are 52 proteins in this network that are expressed in the urinary tract, according to the STRING database, of which 13 are expressed in the human bladder. According to the Reactome pathways database, the top associated pathways were “immune system” (HSA-168256: 93 proteins, *p* = 4.01 × 10^−28^) and “neutrophil degranulation” (HSA-6798695: 46 proteins, *p* = 1.37 × 10^−23^), while according to WikiPathways, among the top associated pathways were “VEGFA-VEGFR2 signaling pathway” (WP3888: 31 proteins, *p* = 2.74 × 10^−05^) and “complement system” (WP2806: 12 proteins, *p* = 1.56 × 10^−06^). According to the STRING database of disease-gene associations, “carcinoma” was among the top associated diseases (DOID:305: 21 proteins, *p* = 9.2 × 10^−3^).

### 2.4. Correlation of Proteomics Findings with Clinical Parameters

Correlation of the proteomics data with clinical records (stage, age) was performed for the 494 proteins with significantly altered abundance among groups. As the data normality was rejected by the normality tests, Spearman’s rho correlation was performed. Significantly correlated with age were 125 proteins. Significantly correlated with stage were 445 proteins, of which, 156 were positively and 289 were negatively correlated. Statistically significant differences in abundance between the groups and correlation with stage with consistent regulation trends (up- or down-) across the BCa stages showed 15 proteins, of which 5 had increased and 10 decreased levels with BCa progression (Figure 3). Nicotinamide N-methyltransferase (NNMT), Galactokinase (GALK1), Serine protease HTRA1 (HTRA1), Complement C1q subcomponent subunit A (C1QA), and Importin-4 (IPO4) showed raise, while Rootletin (CROCC), Alpha-actinin-2 (ACTN2), Tubulointerstitial nephritis antigen-like (TINAGL1), Peptidyl-prolyl cis-trans isomerase FKBP1A (FKBP1A), Polyadenylate-binding protein 4 (PABPC4), Aflatoxin B1 aldehyde reductase member 4 (AKR7L), Erlin-2 (ERLIN2), Septin-9 (SEPTIN9), Intersectin-2 (ITSN2), and S-methylmethionine--homocysteine S-methyltransferase (BHMT2) showed a decrease in the protein abundance with increasing BCa stage.

Statistically significant differences between initial (Ta–T1) and advanced (T2–T3) disease stages showed 14 proteins, of which 7 had increased and 7 decreased levels in advanced BCa (Figure 4). Statistically significant increase in advanced BCa showed ER membrane protein complex subunit 2 (EMC2), UDP-glucuronosyltransferase 1A10 (UGT1A10), Estradiol 17-beta-dehydrogenase 2 (HSD17B2), NSFL1 cofactor p47 (NSFL1C), Leucine zipper putative tumor suppressor 2 (LZTS2), Glycine--tRNA ligase (GARS1) and Fanconi anemia group A protein (FANCA). On the other hand, Electron transfer flavoprotein subunit beta (ETFB), Nucleosome assembly protein 1-like 4 (NAP1L4), Keratin_type I cytoskeletal 20 (KRT20), Plastin-1 (PLS1), Lupus La protein (SSB), Thioredoxin (TXN) and Isocitrate dehydrogenase [NADP] cytoplasmic (IDH1) showed statistically significant decrease in advanced BCa.

### 2.5. Validation of Selected Candidates in Urine

The preselected protein candidates NNMT, GALK1, and HTRA1 were measured in two cohorts of urine samples (Figure 5). The testing in Cohort I showed that all 3 proteins have detectable levels in urine using antibody-based measurements, can discriminate normal controls from BCa patients with statistical significance, and their concentration increases with BCa stage. In addition to discriminating between normal controls and BCa, NNMT showed statistically significant discrimination between initial (Ta, G1) and advanced tumors (T1–T2, G3). As for GALK1 and HTRA1, no significant differences were observed between the BCa stages. The testing in Cohort II validated the findings in regard to NNMT and GALK1 but failed to confirm the HTRA1 differences between BCa and controls. NNMT showed very good statistically significant discrimination between all groups except discrimination between more advanced BCa groups T1 G3 and T2 G3. GALK1 also discriminated most groups with statistical significance, but could not discriminate initial BCa stage (Ta, G1) from normal controls as well as more advanced BCa groups T1 G3 and T2 G3.

It is worth pointing out that the concentration of the tested proteins was 5–10 times higher in the urine samples from Cohort I compared to the samples from Cohort II. This was probably due to the collection method and freezing of the samples without clearing the cell debris, which lysed and released epithelial cell content into the urine. As Cohort II represents the “standard” collection method for urine testing, further calculation in regards to the diagnostic potential in detecting and prognosis of BCa was carried out on samples from Cohort II.

To determine the diagnostic potential in detecting BCa, samples were grouped into two groups: Control and BCa (Ta, G1+ T1–T2, G3). Significant differences in protein levels between Controls and BCa showed NNMT (*p* < 0.0001) and GALK1 (*p* = 0.006) (Figure 6A). The area under curve (AUC) for NNMT and GALK1 were 1.000 (95% CI: 1.000–1.000; *p* < 0.0001) and 0.801 (95% CI: 0.655–0.947; *p* < 0.0001), respectively. The optimal cutoffs for the proteins were: 0.018 ng NNMT/mg creatinine (100% specificity, 100% sensitivity) and 2.88 ng GALK1/mg creatinine (87.5% specificity, 72.7% sensitivity). The diagnostic model based only on GALK1 correctly classified 82.69% of total cases (Control: 0 %; BCa: 97.73%), while the model based only on NNMT correctly classified 100% of total cases (Control: 100%; BCa: 100%).

To determine the diagnostic potential in detecting advanced BCa, samples were grouped into three groups: Control, non-invasive low-grade BCa (Ta, G1), and invasive high-grade BCa (T1–T2, G3). Statistically significant differences between non-invasive low-grade and invasive high-grade BCa showed NNMT (*p* = 0.003) and GALK1 (*p* = 0.001) (Figure 6B). The individual accuracy for NNMT was AUC = 0.763 (95% CI: 0.606–0.921; *p* = 0.001) while for GALK1 AUC was 0.801 (95% CI: 0.653–0.950; *p* < 0.0001). The optimal cutoffs of the proteins were: 1.18 ng NNMT/mg creatinine (75.0% specificity, 82.1% sensitivity) and 3.30 ng GALK1/mg creatinine (75.0% specificity, 85.7% sensitivity). The diagnostic model based only on NNMT correctly classified 72.73% of total cases (Ta, G1: 43.75%; T1–T2, G3: 89.29%), while the model based only on GALK1 correctly classified 77.27% of total cases (Ta, G1: 56.25%; T1–T2, G3: 89.29%). The combination of NNMT and GALK1 slightly increased the diagnostic accuracy to AUC = 0.813 with 77.27% of total correct classification and 56.25% and 89.29% correct classification of non-invasive low-grade BCa and invasive high-grade BCa, respectively.

To investigate whether the urine levels of the biomarkers were clinically independent prognostic factors for BCa, correlation analysis using Spearman rho was performed (Table 2). Both GALK1 and NNMT showed a weak positive correlation with age (rGALK1 = 0.360, *p* = 0.009; rNNMT = 0.351, *p* = 0.011) and moderate positive correlation with stage (rGALK1 = 0.566, *p* < 0.0001; rNNMT = 0.627, *p* < 0.0001). A moderate positive correlation was observed also between GALK1 and NNMT (r = 0.634; *p* < 0.0001).

## 3. Discussion

The high recurrence rate of BCa and the use of invasive diagnostic and monitoring methods require the development of new molecular tools that will introduce non-invasive testing with improved diagnostic accuracy as well as assessment of disease progression and recurrence. Therefore, the identification of new molecular marker(s) that can be used for early detection, particularly for low-grade and low-stage NMIBC as well as risk stratification of BCa patients remains a pertinent clinical need.

To support the implementation of new non-invasive BCa biomarkers into practice, we have set up this comparative proteomics study analyzing the proteome of BCa tissues from initiation to advanced stages. Bioinformatics enrichment analysis of the proteins with altered abundance between stages revealed an association with more than 100 pathways according to the Panther database. The top 20 pathways, that have been preselected to include more than 4 differentially abundant proteins from our dataset, were all strongly associated with cancer, except the “Blood coagulation pathway” (P00011) that showed significant association due to the bleeding and angiogenesis mechanisms associated with BCa. The top 3 associated pathways were “Integrin signaling pathway” (P00034), “Inflammation mediated by chemokine/cytokine” (P00031), and “Cytoskeletal regulation by Rho GTPase” (P00016) with 15, 13, and 10 proteins from our data set, respectively. The main role of integrins is to mediate cell adhesion and transmit mechanical and chemical signals to the cell interior. However, recent studies have revealed that various mechanisms deregulate integrin signaling in cancer, so it can drive multiple stem cell functions, including tumor initiation, epithelial plasticity, metastatic reactivation, and resistance to oncogene- and immune-targeted therapies [16]. Chemokines are soluble factors shown to play important roles in regulating immune cell recruitment during inflammatory responses. While studies in the past have focused solely on the role of chemokine signaling pathways in regulating immune responses, emerging studies show that these molecules regulate diverse cellular processes including angiogenesis, epithelial cell growth, and survival, and as such are critical for cancer progression and direct complexes and diverse functions in the tumor microenvironment [17]. The Rho family of GTPases is highly conserved and contributes to several cellular processes including the organization of the actin and microtubule cytoskeletons, regulation of gene expression, vesicle trafficking, cell cycle progression, cell morphogenesis, cell polarity, and cell migration. However, Rho GTPases also play an important role in cancer. There is growing evidence that in most cancers, expression levels and/or activity of Rho GTPases is altered which puts Rho GTPase signaling as a possible target in the development of new cancer treatments [18]. Moreover, our dataset was significantly associated with several pathways which aberrant regulation represents the hallmark of cancer. These are the “Apoptosis signaling pathway” (P00006) [19], “FAS signaling pathway” (P00020) [20], and “p53 pathway” (P00059) [21] which have a central role in the physiological regulation of programmed cell death and have been implicated in the pathogenesis of various malignancies; “EGF receptor signaling pathway” (P00018) [22], “PI3 kinase pathway” (P00048) [23] and “Cadherin signaling pathway” (P00012) [24] that are one of the most important pathways that regulate growth, proliferation, survival, and differentiation in mammalian cells; “Wnt signaling pathway” (P00057) which is one of the key cascades regulating development and tightly associated with cancer [25] and “Ubiquitin proteasome pathway” (P00060) which plays a pivotal role in the degradation of proteins and has well-established role in the pathogenesis of various human diseases among which is cancer as well [26]. Among the top 20 pathways were “Glycolysis” (P00024) and “Pentose phosphate pathway” (P02762) that branches from glycolysis at the first committed step of glucose metabolism, which have an important function in the regulation of cancer cell metabolism and survival [27,28]; “Heterotrimeric G-protein signaling” (P00026/P00027) that impacts oncogenesis at multiple levels by regulating tumor angiogenesis, immune evasion, metastasis, and drug resistance [29] and “FGF signaling pathway” (P00021) that governs fundamental cellular processes such as cell survival, proliferation, migration, differentiation, is mediated by MAPK and PI3K-AKT pathway, and intersects and synergizes with other signaling pathways such as Wnt, retinoic acid (RA) and transforming growth factor (TGF)-β signaling.

According to the Reactome pathways database, the top associated pathways were “Immune system” (93 proteins) and “neutrophil degranulation” (46 proteins), while according to WikiPathways, among the top associated pathways were “VEGFA-VEGFR2 signaling pathway” (31 proteins) and “complement system” (12 proteins). The role of the immune system in cancer has been extensively studied. In principle, tumor development can be controlled by cytotoxic innate and adaptive immune cells. However, as the tumor develops, cancer cells evolve different mechanisms that mimic peripheral immune tolerance to avoid the tumoricidal attack. The Neutrophil degranulation process plays an important role in introducing new membrane proteins on the surface of neutrophils and dictates interaction between neutrophils and cancer cells, together with other cell populations in the tumor microenvironment. It is well known that neutrophils physically interact with circulating tumor cells, and they can promote tumor progression by stimulating angiogenesis and matrix remodeling and disabling T cell-dependent antitumor immunity [30]. VEGFA signaling through VEGFR2 is the major pathway that activates angiogenesis by inducing the proliferation, survival, sprouting, and migration of endothelial cells (ECs), and also by increasing endothelial permeability [31]. As for the complement system, in a tumor context, it may affect the immunity, angiogenesis, and phenotype of the tumor cells while in terms of immunity, the same complement proteins may influence several immune cells positively or negatively depending on the model or cancer type [32]. 

Further, a comparison of our dataset of differentially abundant proteins with the most promising tissue protein biomarkers for BCa diagnosis/prognosis from published studies (reviewed in [12,15]) showed that our dataset contains a number of these proteins. Namely, our dataset contained Alpha-actinin-1 (ACTN1), Matrix metalloproteinase-9 (MMP9), Thymidine phosphorylase (TYMP), Cullin-associated NEDD8-dissociated protein 1 (CAND1), Beta-hexosaminidase subunit beta (HEXB), Alpha-internexin (INA), Small ribosomal subunit protein eS19 (RPS19) reported to be BCa specific [33] and candidate biomarkers for advanced stage BCa such as Exocyst complex component 4 (EXOC4), MMP9 [34], Cathepsin E (CTSE) [35], L-lactate dehydrogenase (LDHB) [36], Fibrinogen Beta (FGB) and Lamin-B1 (LMNB1) [37]. Comparison with potential biomarkers in urine ((reviewed in [11,12,15]) showed several biomarkers that have been proposed as urine biomarkers for BCa multiple times, such as MMP9, FGB, Serine protease HTRA1 and Nuclear mitotic apparatus protein 1 (NUMA1). Worth pointing out is that two FDA-approved tests, namely, ALERE NMP22^®^ TEST and the NMP22™ BLADDERCHEK™ (Abbott Molecular Diagnostics, USA) are based on NUMA1 for BCa detection and surveillance in urine samples [38]. The identification of the above-mentioned biomarkers in our dataset of differentially abundant proteins gives further validation to these proteins as well as confirms the validity of our approach.

Correlation of the proteomics findings with clinical data revealed 15 proteins that showed a high statistically significant correlation with BCa stage and constant up or down trend with disease stages. Out of these, and based on the available literature data about their involvement in cancer and BCa, and positive correlation with stage, we have preselected NNMT, GALK1, and HTRA1 to be further tested in urine samples. 

Nicotinamide N-methyltransferase (NNMT) is an enzyme that catalyzes the N-methylation reaction of nicotinamide, using S-adenosyl-L-methionine as the methyl donor and therefore plays a central role in regulating cellular methylation potential, mainly expressed in the liver and belongs to phase II metabolizing enzymes [39]. NNMT overexpression has been reported for many solid tumors, including gastric, colon, lung, breast, endometrial, cervical, ovarian, oral, esophageal, nasopharyngeal, and thyroid cancers, as well as in epithelial neoplasms (extensively reviewed in [40]). The analysis of NNMT expression levels in different cancers from The Cancer Genome Atlas (TCGA) dataset indicates that NNMT might be a potential biomarker and therapeutic target in some cancers [41]. The recent proteomics work analyzing NNTM in the tumor and stromal compartments of several cancers revealed that this enzyme is a central metabolic regulator of cancer-associated fibroblast differentiation and cancer progression in the stroma that may be therapeutically targeted [42]. These findings have initiated the development of NNMT inhibitors in recent years, starting from varieties of bisubstrate inhibitors [43,44] to structurally diverse NNMT inhibitors such as macrocyclic peptides which bind to NNMT [45]. Overexpression of NNMT was reported in urological cancers as well [40]. In terms of BCa, there are only three studies that have reported the possible biomarker role of NNMT and these are gene expression studies. The first study that aimed to investigate the differentially expressed genes in relation to the BCa stages revealed that NNMT was significantly upregulated in MIBC compared to NMIBC [46]. The subsequent study, based on gene expression cDNA macroarray data, confirmed the overexpression in BCa tissues compared to controls and validated the findings by qPCR, Western blot, and catalytic activity assay [47]. This study also reported that NNMT expression levels in urine were significantly higher in BCa patients compared to controls which showed low or undetectable amounts of NNMT transcript and protein. In the latest study, using qPCR, NNMT expression in urine samples from BCa patients was significantly higher compared to controls, but inversely correlated with histological grade [48]. Our study accessed the protein level of NNMT in BCa tissues for the first time. The results from the comparative proteomics study showed that the protein level of NNMT in BCa tissues increased steadily with stages and that based on the tissue protein levels, NNMT could discriminate initial, intermediate, and advanced BCa stages with statistical significance. Testing of the protein levels in urine confirmed NNMT steady increase with BCa stages with very good discrimination between controls, initial, intermediate, and advanced stages, excellent diagnostic, and solid prognostic potential. Overall, our study showed that protein levels of NNMT in both tissue and urine correspond to the reported gene expression [46,47] and confirm NNMT as a potential biomarker for both BCa diagnosis and prognosis. However, the results in regards to NNMT correlation with BCa stage were opposite to the previous findings on mRNA levels [48].

Galactokinase (GALK1) is a major enzyme for the metabolism of galactose that catalyzes the transfer of a phosphate from ATP to alpha-D-galactose and participates in the first committed step in the catabolism of galactose. It is a ubiquitously expressed enzyme with the highest expression in the liver. GALK1’s relation with cancer was first observed by Barretina and co-workers [49] who reported that the GALK1 gene was up-regulated by at least six-fold in 28 different human liver cancer cell lines. The subsequent study applying small interfering RNA (siRNA) to target the GALK1 gene in the hepatocellular carcinoma (HCC) cell line supported GALK1 as a novel target for treating HCC and uncovered new posttranscriptional regulatory mechanisms that link the galactose metabolic pathway to protein expression of the PI3K/AKT pathway [50]. Very recently, in the last two years, GALK1 was linked also to BCa. Based on the well-established knowledge that energy metabolism and its reprogramming is an essential hallmark of most cancers, mRNA expression profiling of glycolysis-related genes [51], metabolism-related genes [52], and energy metabolism-related genes [53] in BCa cohorts by mining data from The Cancer Genome Atlas (TCGA) database was performed. GALK1 was part of the four gene panel [51], 16 gene panel [52], and 13 gene panel [53] that could be used for BCa prognosis. The expression level of GALK increased with increasing risk score and was inversely correlated with the patient’s survival [53]. The results from our comparative proteomics study showed that the protein level of GALK1 in BCa tissues increased steadily with stages and that based on the tissue protein levels, GALK1 could discriminate initial, intermediate, and advanced BCa stages with statistical significance. The diagnostic potential in urine was characterized with solid accuracy but lower than NNMT, while prognostic accuracy was the highest with the correct classification of 77.3% of cases.

Serine protease HTRA1 is a ubiquitously expressed protein, a member of the trypsin family of serine proteases that is proposed to regulate the availability of insulin-like growth factors (IGFs) and has also been suggested to be a regulator of cell growth. HTRA1 is involved in several vascular diseases and its altered expression has been reported for a few cancers such as ovarian [54], lung cancer [55], and melanoma [56], where a tumor suppressor role was proposed. Dysregulation concerning BCa has been reported only in one study, where tissue levels of HTRA1 were undetectable in a few urothelial cancer cell lines but significantly higher amounts were found in urine from cancer patients compared with both healthy subjects and patients with cystitis [57]. The results from our study are opposite to previous findings in terms of tissue expression in the above-mentioned cancers. According to the detected levels in BCa tissues, HTRA1 increased steadily with stages. Recent study where inhibition of HTRA1 in the tumor stroma impaired tumor progression by deregulating angiogenesis [58], is in favor of HTRA1 acting as an oncogene and in concordance with our findings. Testing in the first urine cohort showed that HTRA1 could discriminate between controls and BCa patients and a trend of increasing protein levels with BCa stages was observed as in the study of Lorenzi et al., [57]. However, testing in the second urine cohort failed to confirm the HTRA1 differences between BCa and controls. Overall, more work is needed to establish the role of HTRA1 in cancer in terms of whether it functions as an oncogene or a tumor suppressor. However according to our results, its potential as a biomarker in BCa is low; more studies are necessary to evaluate its potential. 

In addition to the above potential biomarkers, our list contains several more proteins that showed high statistically significant positive or negative correlation with BCa stage with steady increase or decrease, respectively, and according to HPA are associated with some cancers. To name a few, Importin-4 (IPO4), Peptidyl-prolyl cis-trans isomerase FKBP1A (FKBP1A), and Polyadenylate-binding protein 4 (PABPC4) are prognostic biomarkers for liver and renal cancer; Complement C1q subcomponent subunit A (C1QA) is a prognostic biomarker for renal cancer; Erlin-2 (ERLIN2) is a prognostic biomarker for glioma and renal cancer; Intersectin-2 (ITSN2) is a prognostic biomarker for head and neck cancer; Tubulointerstitial nephritis antigen-like (TINAGL1) is increased in urothelial cancers and is prognostic biomarker for renal and thyroid cancers; Septin-9 (SEPTIN9) is a prognostic biomarker for liver cancer; Rootletin (CROCC) is a prognostic biomarker for lung and renal cancer. Validation of some of these proteins in the context of BCa might bring more potential biomarkers for this disease.

It is worth mentioning that this study, besides its reliable design, technical approach, and high-quality bioinformatics and statistical analysis possesses some limitations. One of the limitations is the relatively small sample cohorts used for discovery and validation. Although we have available biobank with considerable number of samples, due to the often very low amounts of tissue available from Ta tumors with Grade 1, we were restricted to perform the comparative proteomics analysis with 6 samples per group, or 18 samples in total. Urine cohorts were restricted in size since the large proportion of samples had visible hematuria which interfered with ELISA results. Other limitations include variations in urine storage time and conditions before the delivery to our lab, as well as inherited variability of the chosen technique for measurement of the proteins in urine.

In order to overcome these limitations and establish more precisely the value of the proposed biomarkers for non-invasive BCa detection and prognosis, several aspects need to be included in future studies. First, validation of NNMT, GALK1, and HTRA1 and other candidates tightly correlated with BCa stage needs to be performed using larger patient cohorts of BCa including benign conditions such as cystitis, samples with hematuria, as well as other tumors from the urogenital tract (prostate, and renal). Inclusion of samples from other urogenital cancers is of particular importance as the identified and validated biomarkers in this study are not bladder-specific but ubiquitously expressed proteins which are often dysregulated in cancer. Second, assessment of the protein intra- and inter-variability in urine is a critical step toward clinical application. Third, comparison of the ELISA results with other methods for precise quantification, preferably targeted proteomics. This would be necessary in particularly for urine samples with hematuria which are prevalent in BCa patients. The findings from this study also open the way for testing the transcription levels of the proposed protein candidates in tissue and urine, which could lead to the development of more rapid and cost-effective test for BCa prognosis.

## 4. Materials and Methods

### 4.1. Patients Samples

The tissue samples used for the proteomics profiling are part of the RCGEB MASA biobank of snap-frozen tumors from patients with BCa. This biobank currently includes tissue and/or urine samples from approximately 170 patients. The diagnosis of the patients was based on histological evaluation of the tissues obtained by surgical procedure. The biobank contains tissue and urine samples from all stages of BCa with tumor sizes from Ta to T3, classified into Grade 1 (*n* = 33), Grade 2 (*n* = 78), and Grade 3 (*n* = 61). The study was designed to investigate proteomics alterations in BCa progression starting from low-grade (G1) Ta tumors to high-grade (G2–G3) T1–T2 tumors. For the comparative proteomics analysis by label-free data-independent LC-MS/MS, a total of 18 tissue samples were profiled, grouped into 3 groups of 6 samples each: (1) Group 1 (Ta, G1), (2) Group 2 (T1, G2–G3) and (3) Group 3 (T2–T3, G3). Patients were aged 39–82 years with no significant differences among groups regarding age (Table 3, Appendix A).

The selected potential biomarkers from the comparative tissue proteomics analysis were tested in urine samples from BCa patients and controls. We had two cohorts of urine samples: Cohort I consisted of samples taken during the cystoscopy and frozen immediately at −80 °C without any processing; and Cohort II consisted of urine samples collected according to standard guidelines as first-morning urine, that were further centrifuged at 1000× g for 10 min to remove cell debris, aliquoted in 1.5 mL tubes, and stored at −80 °C until use. As a first criterion, we preselected the urine samples, excluding samples with hematuria. Although the initial aim was for Cohort I to include urine samples from the same patients that were included in the comparative tissue proteomics analysis, in the end only 5 urine samples were included from these patients, due to the presence of hematuria in the remaining samples. The remaining samples that had no hematuria were from other patients. Currently, the biobank contains a limited number of urine samples collected during the cystoscopy we had only 24 urine samples without hematuria, divided into the following groups: Group 1 (Ta, G1) (*n* = 9); Group 2 (Ta, G2) (*n* = 9); Group 3 (T1, G3) (*n* = 4) and Group 4 (T2, G2–G3) (*n* = 2). Cohort II consisted of 52 urine samples of which, 8 controls without BCa and 44 BCa samples without hematuria were divided into the following groups: Group 1 (Ta, G1) (*n* = 16); Group 2 (T1, G2–G3) (*n* = 13); and Group 3 (T2, G3) (*n* = 15). There were no significant differences among groups regarding age in both urine cohorts (Table 3, Appendix A).

### 4.2. Sample Preparation

Protein extraction from the fresh frozen tissues (10–15 mg per sample) was pulverized with liquid nitrogen. Tissue powder was dispersed in Lysis buffer (4% SDS, 5 mM MgCl_2_x6H_2_O, 10 mM CHAPS, 100 mM NH_4_HCO_3_, 50 mM DTT) in a 1:20 ratio (*w*/*v*), mixed and allowed to dissolve by sonication in an ice bath for 30 min. If the samples were viscous, Lysis buffer was added to max. 10% of the starting volume and the samples were vortexed and sonicated again. The protein content was quantified by the Bradford method [59]. Samples were prepared for LC-MS/MS using RapiGest [60] as previously described in detail [61].

### 4.3. LC-MS/MS Data Acquisition

A label-free LC-MS/MS protein profiling was performed using ACQUITY UPLC^®^ M-Class (Waters Corporation, Milford, MA, USA) coupled with SYNAPT G2-Si High Definition Mass Spectrometer (Waters Corporation, Milford, MA, USA). Data were obtained using ion-mobility separation (IMS) acquisition named ultradefinition MS^E^ (UDMS^E^) [62]. Optimization and determination of the optimal peptide load on column was performed using pool sample, containing an equal amount of each 18 individual samples. The pool sample was run from 100–400 ng per run. The determined optimal concentration was 400 ng per run. Each sample had one test run and initial data processing with ProteinLynx Global SERVER (PLGS, version 3.0.3, Waters Corporation, Milford, MA, USA) for quality assurance, followed by a run at the optimal concentration.

Peptides were trapped on an ACQUITY UPLC M-Class Trap column Symmetry C18, 5 µm particles, 180 µm × 20 mm, (Waters Corporation), followed by separation on ACQUITY UPLC M-Class reverse phase C18 column HSS T3, 1.8 µm, 75 µm × 250 mm (Waters Corporation, Milford, MA, US) at a flow rate of 300 nL/min using 90 min multistep concave gradient [63]. Lock mass compound Glu-1-Fibrinopeptide B (EGVNDNEEGFFSAR) with a concentration of 100 fmol/µL was delivered by the auxiliary pump of the LC system at 500 nL/min, every 45 s. Spectra were recorded in resolution positive ion mode. Mass spectrometric settings were as previously described in detail [61].

### 4.4. LC-MS/MS Data Processing

Data was searched against the UniProtKB/Swiss-Prot database containing 20,370 proteins (June 2020), with added yeast alcohol dehydrogenase (UniProt P00330) sequence. Test runs were processed using PLGS (Waters Corporation) with the following settings: (1) low energy (LE) and high energy (HE) threshold settings of 150 counts and 30 counts, respectively; (2) Precursor and fragment ion mass tolerances set to auto; (3) Search settings included one missed cleavages, carbamidomethyl C as a fixed modification, and oxidized M as a variable modification; (4) A minimum of two fragment ion matches was required per peptide identification and five fragment ion matches per protein identification, with at least one peptide match per protein identification; (5) The protein false discovery rate (FDR) was set to a 1%; (6) internal standard protein, P00330 with concentration of 25 fmol/μL. The data were post-acquisition lock mass corrected using the doubly charged monoisotopic ion of [Glu1]-Fibrinopeptide B. The typical range of RMS error for precursor and product ions for were ±5 and ±10 ppm, respectively.

Comparative proteomics analysis was carried out using Progenesis QIP version 4.1 (Nonlinear dynamics, Waters Corporation). The following settings were applied: (1) LE and HE threshold set to auto; (2) reference run-auto; (3) normalization-“normalize to all proteins”; (4) digest reagent-trypsin; (5) maximum missed cleavages-one; (6) maximum protein mass-250 kDa; (7) fixed modifications–carbamidomethyl C; (8) variable modification–oxidation M; (9) peptide tolerance-auto; (10) fragment tolerance-auto; (11) FDR < 1%; (12) Ion matching requirements as in PLGS processing; (13) Quantification based on non-conflicting peptides; (14) Grouping of similar proteins; (15) The combined target-decoy database for database search. Data were further filtrated to remove peptides with a sequence length of less than six amino acids and a score below 4. Proteins and peptides were exported in the form of a .csv output files for subsequent data analysis. The calculated FDR on the whole dataset level was 3.68%.

### 4.5. Quantitative Measurement of Candidate Proteins in Urine

For the quantitative measurement of the selected proteins in urine, we used the following ELISA kits: Human Nicotinamide-N-Methyltransferase (NNMT) ELISA (Cat. No.: MBS453001) with a lower limit of detection (LLD) = 0.059 ng/mL; Human Galactokinase 1 (GALK1) ELISA Kit (Cat. No.: MBS8804227) with LLD = 1.57 ng/mL; and Human Serine protease HTRA1 (HTRA1) ELISA Kit (Cat. No.: MBS902316) with LLD = 0.39 ng/mL. ELISA kits were purchased from MyBioSource.com. Samples were assayed using 100 µL undiluted urine, in duplicate, according to the manufacturer’s instructions. The concentrations of NNMT, GALK1, and HTRA1 were normalized to urine creatinine to correct for variations in urinary concentration.

### 4.6. Data Analysis

Differentially abundant proteins were selected based on Anova ≤ 0.05. Statistically significant protein levels between groups were determined by the Mann–Whitney U test and corrected using the Benjamini-Hochberg procedure [64]. Panther [65] and STRING [66] databases were used for functional annotation and enrichment analysis, respectively. STRING settings included (1) full STRING network; (2) evidence setting; (3) all active interaction sources; (4) medium confidence score; and (5) max number of interactors to show, for the 1st shell-none/query proteins only, and for the 2nd shell-none. The tissue specificity and distribution of the selected proteins in the normal bladder tissue and in cancer was evaluated based on the mRNA expression data from Human Protein Atlas version 21.0 [67].

Statistical analyses included: (1) Shapiro–Wilk, Anderson-Darling, Lilliefors, and Jarque-Bera tests for data distribution; (2) Mann–Whitney U-test for two-sample comparisons; (3) Spearman’s rho correlation for the correlation of the quantitative proteomics data with the patient’s clinical and histopathological reports; (4) logistic regression with clinical diagnosis or cancer stage as the dependent variable and protein concentrations as independent variables; and (5) Receiver operating characteristic (ROC) curves. Diagnostic performance was defined by area under the curve (AUC). A confidence level of 95% (*p* < 0.05) was considered significant for all performed tests. These tests were performed using XLSTAT software ver. 2022.1.2 (https://www.xlstat.com).

## 5. Conclusions

Using comparative proteomics approach in analyzing BCa tissues ranging from initial to advanced stages, we have identified pathways that are involved in BCa pathogenesis and panel of potential biomarker candidates that displayed strong correlation with the progression of the disease. We chose to validate three candidate biomarkers, NNMT, GALK1, and HTRA1, for which limited data about their relationship with BCa, based mainly on gene expression and IHC methods, was available. Validation revealed a clinically relevant correlation between tissue and urine concentrations of NNMT and GALK1 with BCa. Our study for the first time accessed the protein levels of NNMT and GALK1 in BCa patients’ tissues and urine as well as reported their correlation with disease stage. The observed relationship proposes two new biomarkers for non-invasive diagnosis and prognosis of BCa. In addition, this study opens a way to further testing and validation of more high-quality proteomics biomarkers that could ultimately add value to the clinical management of BCa.

## Figures and Tables

**Figure 1 ijms-24-14938-f001:**
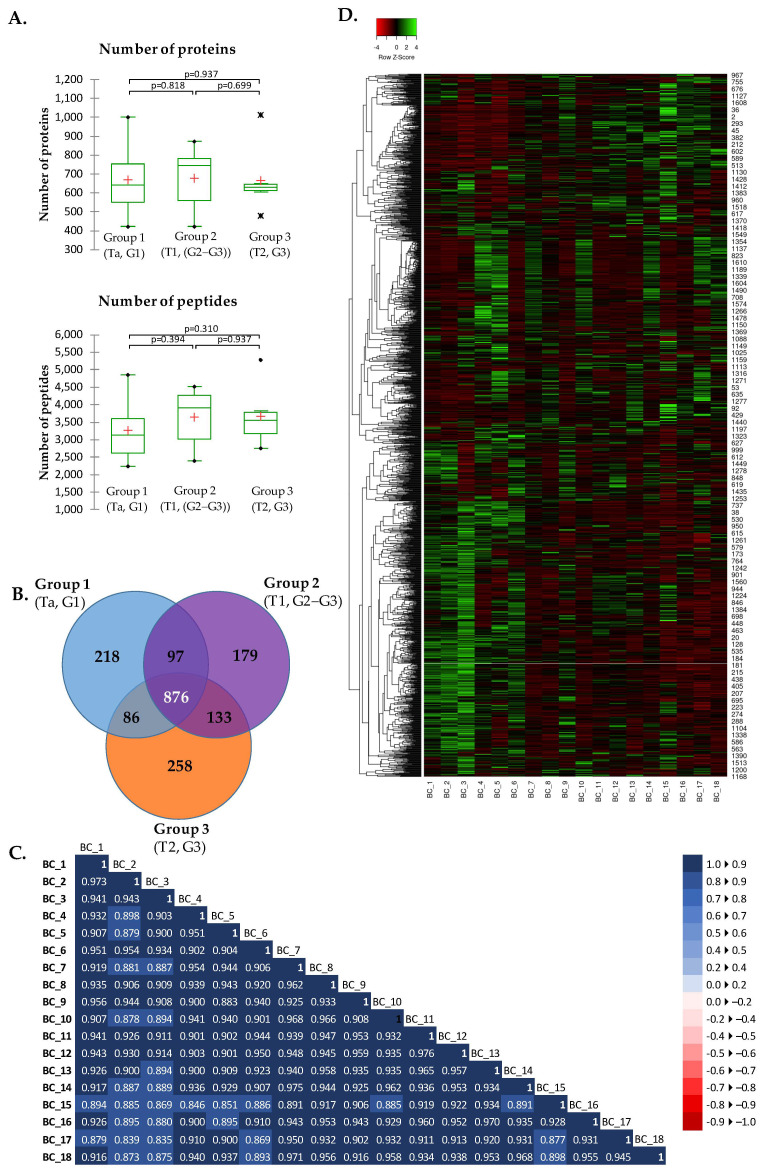
Summary of the proteomics data identified and quantified by Progenesis QIP. (**A**) Number of proteins and peptides identified in the experimental groups. In the box plot graphs, median (−), 25th and 75th percentiles, minimum/maximum (•) mean (+) and outliers (∗) are shown. (**B**) Number of unique and shared proteins among groups (**C**) Correlation matrix for normalized protein abundances across the individual samples using Spearman Rho correlation. (**D**) Heatmap of the normalized protein abundances of 1847 proteins where samples are shown in columns and proteins in the rows. Clustering method applied: Average linkage; Distance Measurement method: Spearman rank correlation.

**Figure 2 ijms-24-14938-f002:**
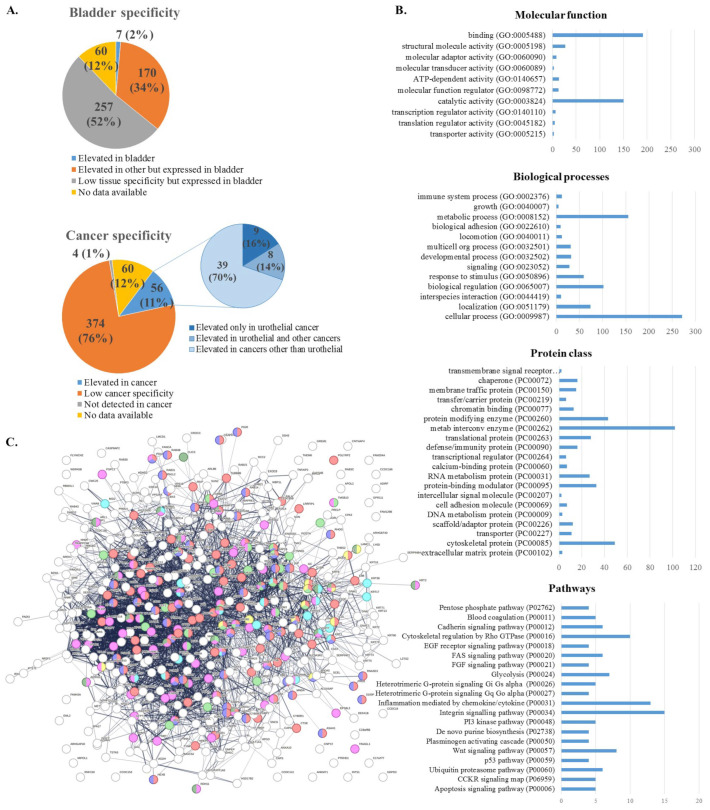
Categorization of the differentially abundant proteins in BCa. (**A**) Bladder tissue and cancer specificity according to the data from the Human Protein Atlas, based on transcript mRNA detection and specificity. (**B**) GO annotations of proteins according to the Panther Classification System. (**C**) Enrichment analysis with STRING: Proteins colored in pink are expressed in the urinary system, while proteins colored in dark green are expressed in the human bladder. The top associated pathways according to the Reactome pathways database were the immune system (red) and neutrophil degranulation (blue), while according to WikiPathways, these were the VEGFA-VEGFR2 signaling pathway (green) and complement system (yellow). Carcinoma was among the top associated diseases (turquoise) according to the STRING database of disease-gene associations.

**Figure 3 ijms-24-14938-f003:**
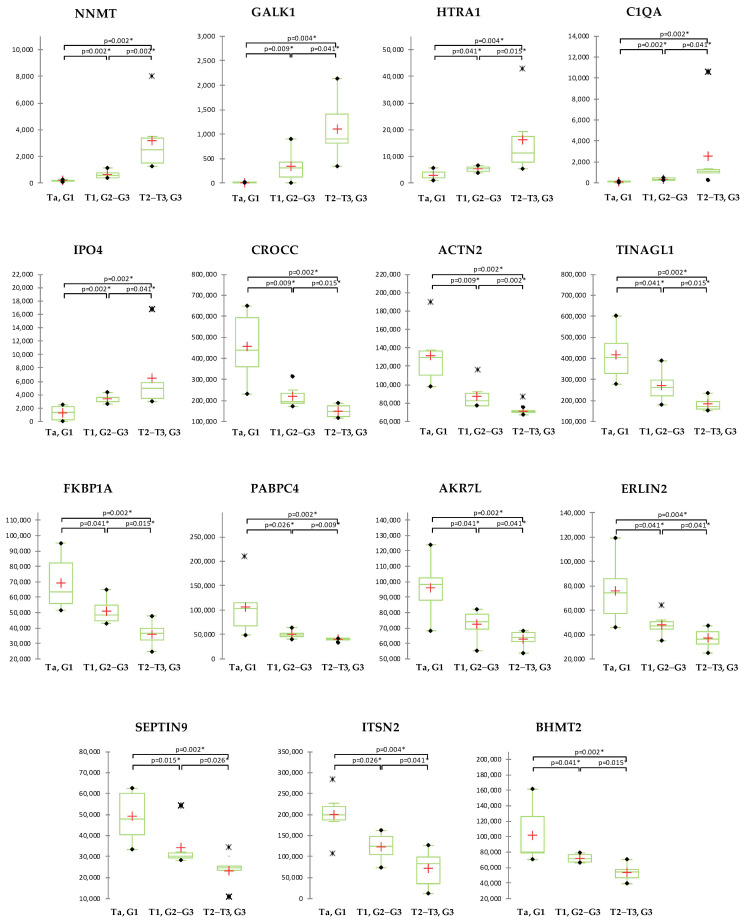
Abundance of the proteins significantly correlated with cancer stage with steady up or down with disease stages. All of the presented proteins showed statistically significant differences in abundance between cancer groups (* *p* < 0.05). In the box plot graphs, median (−), 25th and 75th percentiles, minimum/maximum (•), outliers (∗) and mean (+) are shown.

**Figure 4 ijms-24-14938-f004:**
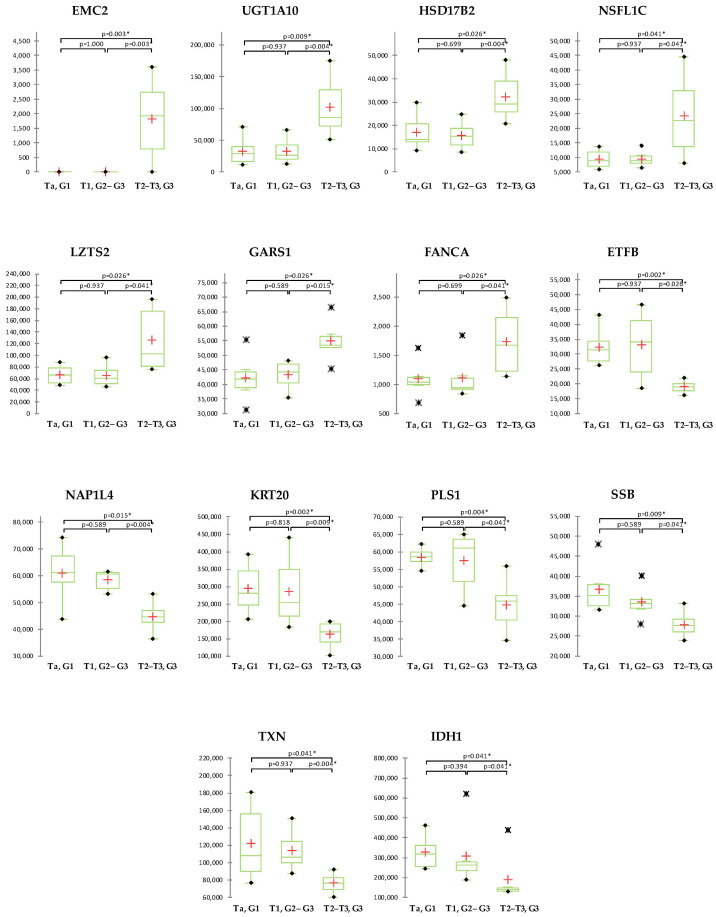
Abundance of the proteins significantly correlated with cancer stage with statistically significant difference in protein abundance between initial (Ta/T1) and advanced (T2/T3) tumors. Significant differences between investigated cancer groups are marked (* *p* < 0.05). In the box plot graphs, median (−), 25th and 75th percentiles, minimum/maximum (•), outliers (∗) and mean (+) are shown.

**Figure 5 ijms-24-14938-f005:**
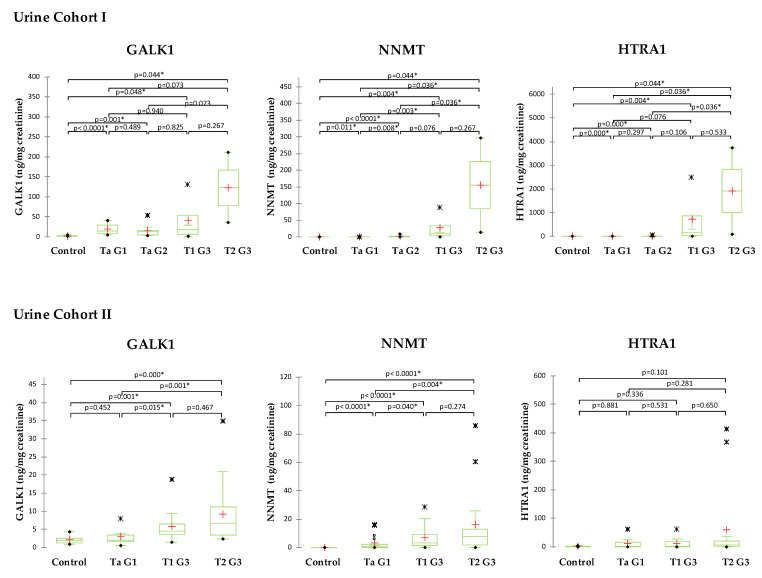
Normalized protein levels of the 3 validated proteins in urine samples of BCa patients and controls. Proteins were validated in two cohorts of urine samples: Cohort I consisted of 24 samples BCa samples taken during the cystoscopy and frozen immediately at −80 °C without any processing; and Cohort II consisted of 44 urine samples from BCa patients collected according to standard guidelines as first-morning urine, centrifuged, aliquoted and stored at −80 °C. Control group consisted of 8 samples from individuals without BCa. Statistically significant differences accessed by the Mann–Whitney U-test are marked (* *p* < 0.05). In the box plot graphs, median (−), 25th and 75th percentiles, minimum/maximum (•), outliers (∗) and mean (+) are presented.

**Figure 6 ijms-24-14938-f006:**
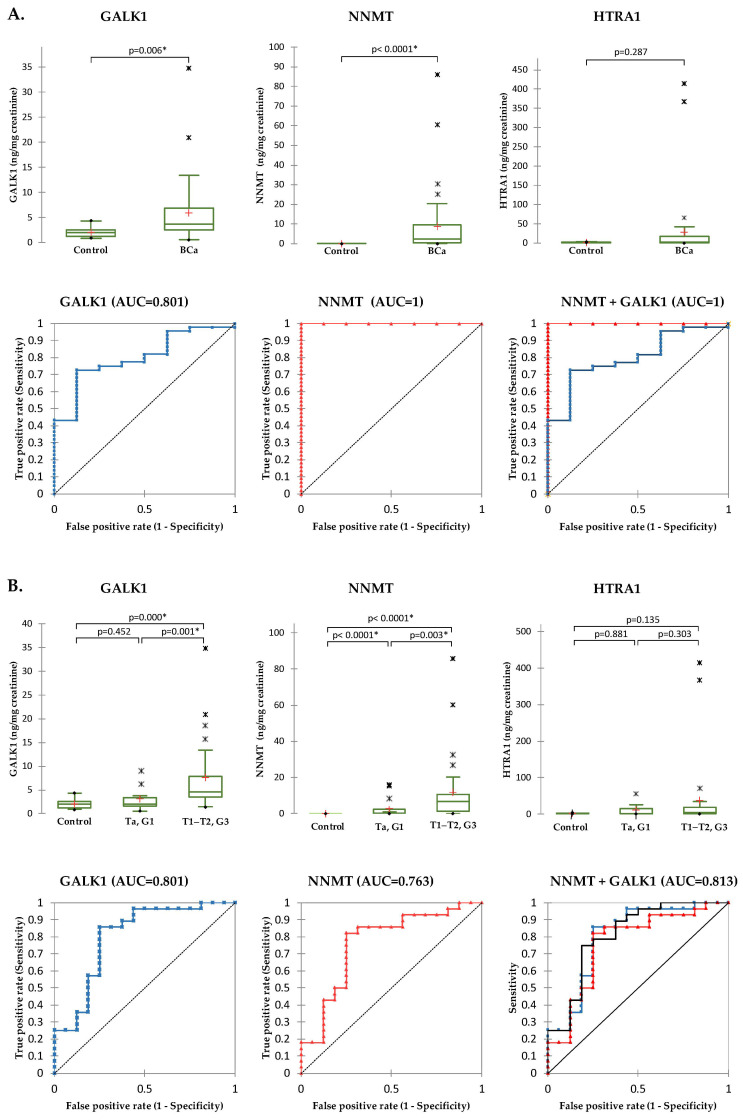
Diagnostic potential of GALK1, NNMT and HTRA1 for BCa diagnosis and prognosis. (**A**) Diagnostic accuracy for early detection of BCa. (**B**) Diagnostic accuracy for discrimination between non-invasive low-grade and invasive high-grade BCa. Normalized protein levels in urine of BCa patients and controls is represented in box plots while the diagnostic accuracy is depicted with receiver operating characteristic (ROC) curves as area under the curve (AUC). In the box plot graphs, median (−), 25th and 75th percentiles, minimum/maximum (•), outliers (∗) and mean (+) are presented. Statistically significant differences are marked (* *p* < 0.05).

**Table 1 ijms-24-14938-t001:** Overview of the number of identified proteins with differential abundance between groups and their regulation trend.

	Gr1 vs. Gr2 (Ta vs. T1)	Gr2 vs. Gr3 (T1 vs. T2/3)	Gr1 vs. GR3 (Ta vs. T2/3)
Differentially expressed (Mann–Whitney *p* ≤ 0.05)	340	95	482
Differentially expressed (B-H *p* ≤ 0.05)	81	0	195
Up-regulated (B-H *p* ≤ 0.05)	35	/	75
Down-regulated (B-H *p* ≤ 0.05)	46	/	121

**Table 2 ijms-24-14938-t002:** Spearman correlation between tested biomarkers in urine and clinical parameters.

Variables	Age	Stage	GALK1	NNMT	HTRA1
Age	**1**	**0.412**	**0.360**	**0.351**	0.188
Stage	**0.412**	**1**	**0.566**	**0.627**	0.206
GALK1	**0.360**	**0.566**	**1**	**0.634**	**0.565**
NNMT	**0.351**	**0.627**	**0.634**	**1**	**0.598**
HTRA1	0.188	0.206	**0.565**	**0.598**	**1**

Values in bold are different from 0 with a significance level of alpha = 0.05.

**Table 3 ijms-24-14938-t003:** Summary of the clinical and histopathological data of patients included in the study.

Group	Diagnosis	Patients Per Group	Age (Mean ± SD)	Age (Median)	TNM Classification			Grade
					T	N	M	
Tissue samples for the discovery of proteomics								
Group 1 (Ta, G1)	Non-invasive low-grade papillary urothelial carcinoma	6	58.5 ± 9.8	62	Ta	N0	M0	I
Group 2 (T1, G2–G3)	Invasive low/high-grade papillary urothelial carcinoma	6	62.2 ± 6.7	62	T1	N0	M0	II–III
Group 3 (T2–T3, G3)	Invasive/Infiltrative high-grade papillary urothelial carcinoma	6	72.2 ± 8.0	73.5	T2–T3	Nx	Mx	III
Urine cohort I for ELISA validation								
Group 1 (Ta, G1)	Non-invasive low-grade papillary urothelial carcinoma	9	58.2 ± 12.0	63	Ta	N0	M0	I
Group 2 (Ta, G2)	Non-invasive low-grade papillary urothelial carcinoma	9	69.9 ± 6.6	68	Ta	N0	M0	II
Group 3 (T1, G3)	Invasive high-grade papillary urothelial carcinoma	4	69.5 ± 1.3	69.5	T1	N0	M0	III
Group 4 (T2, G2–G3)	Infiltrative high-grade papillary urothelial carcinoma	2	76.0 ± 7.1	76	T2	Nx	Mx	II–III
Urine cohort II for ELISA validation								
Group 1 (Ta, G1)	Non-invasive low-grade papillary urothelial carcinoma	16	64.6 ± 14.3	65	Ta	N0	M0	I
Group 2 (T1, G2–G3)	Invasive low/high-grade papillary urothelial carcinoma	13	64.6 ± 7.8	64	T1	N0	M0	II–III
Group 3 (T2, G3)	Invasive/Infiltrative high-grade papillary urothelial carcinoma	15	68.9 ± 10.3	66	T2	Nx	Mx	III
Control group	/	8	48.9 ± 9.8	47	/	/	/	/

## Data Availability

The mass spectrometry proteomics data have been deposited to the ProteomeXchange Consortium via the PRIDE partner repository with the dataset identifier PXD044432 and DOI: 10.6019/PXD044432.

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
