# Peer review of "Proteomics Profiling of Bladder Cancer Tissues from Early to Advanced Stages Reveals NNMT and GALK1 as Biomarkers for Early Detection and Prognosis of BCa"

_ijms, 2023, doi:10.3390/ijms241914938_

Round 1
Reviewer 1 Report
The manuscript “Proteomics profiling of bladder cancer tissues from early to advanced stages reveals NNMT and GALK1 as biomarkers for early detection and prognosis of BCa” is a research article that screened potential novel biomarkers for non-invasive bladder cancer diagnosis and prognosis.
This work is surely interesting for the readers and the design of the study is appropriate. However, several flaws are present which do not allow to accept the manuscript for publication in this form. Authors are strongly encouraged to improve the manuscript accordingly.
1. The manuscript needs a language revision since there are many typos and mistakes.
2. How authors ended up with the sample of BCa subtypes? How many specimens were selected? How many were excluded? What were the reasons for exclusions? Consider presenting a flowchart showing the sampling method.
3. In the discussion section, authors correctly state that NNMT has been proposed also as therapeutic target. In this regard, it is noteworthy to mention that several NNMT inhibitors are already available and ready to be tested also in this malignancy (PMID: 34572571; PMID: 34704059; PMID: 34424711).
4. Authors should include in the main text (summarized from the supplementary materials table 3) a short table summarizing the main clinical parameters of the patients groups. Please also include the mean age.
5. Paragraph “4.2. Sample preparation” contains several mistakes. Chemical formula are all written without subscripts. Please fix it.
6. Line 536: I do not understand why there is double “((“ for manufacturer details.
7. Do the authors have chance to confirm ELISA data with a western blot?
8. Please clearly state the limitations of this study in the discussion section.
The manuscript needs a language revision since there are many typos and mistakes.
Author Response
Dear reviewer,
We have found your comments and suggestions very helpful in the further improvement of the manuscript. Most of the suggestions and questions resulted with the appropriate changes in the manuscript. All changes inserted into the revised version of the manuscript are in red.
Please find below detailed answers to all of the raised points.
Sincerely,
Dr. Katarina Davalieva
Comments by the Reviewer 1
- The manuscript needs a language revision since there are many typos and mistakes.
Thank you for this remark. We have thoroughly checked the text and corrected the spotted grammatical/spelling errors.
- How authors ended up with the sample of BCa subtypes? How many specimens were selected? How many were excluded? What were the reasons for exclusions? Consider presenting a flowchart showing the sampling method.
Thank you for this constructive suggestion. In the section 4.1, we have inserted detailed description of the available sample resources from our BCa biobank as well as how the samples were preselected and grouped in the comparative tissue proteomics analysis and subsequent validation in urine using ELISA (page 16-17). We hope that the present explanation sufficiently depicts the selection process.
- In the discussion section, authors correctly state that NNMT has been proposed also as therapeutic target. In this regard, it is noteworthy to mention that several NNMT inhibitors are already available and ready to be tested also in this malignancy (PMID: 34572571; PMID: 34704059; PMID: 34424711).
Thank you for this suggestion. We have incorporated this information together with references in the discussion (page 14, lines 405-407).
- Authors should include in the main text (summarized from the supplementary materials table 3) a short table summarizing the main clinical parameters of the patients groups. Please also include the mean age.
Thank you for this constructive suggestion. We have summarized the clinical and histopathological data of patients included in the study into a new Table 3 (page 17).
- Paragraph “4.2. Sample preparation” contains several mistakes. Chemical formula are all written without subscripts. Please fix it.
Thank you for this remark. We have corrected chemical formulas.
- Do the authors have chance to confirm ELISA data with a western blot?
No, not yet. Please note that this is a preliminary study where we have identified a panel of proteins that show significant quantitative changes in initial and advanced stages of BCa. As one of the study aims was to test if we can identify biomarkers for non-invasive testing, we chose to validate the putative biomarkers in urine. However, we plan to investigate more profoundly the findings from this study. The future plans include further testing and validation of NNMT, GALK1 and several other identified putative biomarkers on tissue level using Western blot and in urine using ELISA as well as targeted proteomics (we plan to introduce targeted proteomics in our lab in the near future). We also plan to investigate if transcription levels of these biomarkers in tissue and urine correlate with protein levels or not, which could possibly lead to development of non-invasive genetics test. We have included these informations in the last paragraph of the discussion.
- Please clearly state the limitations of this study in the discussion section.
Thank you for this suggestion. We have detailed all the limitations of this study in a more straightforward manner (page 16, lines 490-498). We hope that this aspect is clearer now.
Reviewer 2 Report
This is a well written paper employing LC-MSMS to examine proteins (peptides) that are expressed in bladder cancers as contrasted with normal bladder. The authors present a variety of proteins expressed in cancers vs normal tissues, invasive cancers vs non invasive cancers, and at least two subgroupings of invasive cancers. They then show that certain of the proteins differentiating non invasive and invasive cancers could be observed in the urines of individuals with cancer. The data looks relatively striking and shows that the urine approach can discern these differences. Although as stated by the authors the total numbers of samples employed is limited.
The reviewer feels that there might be additional useful insights that might be gained by a comparison of the present data with the more commonly employed genomic approaches to bladder cancer. Firs using both their primary candidates NNMT, Galk-1 and HTRA are the differences observed in the urine and tissue at the protein level reflected in the RNA expression level. One might wish to examine 10 or 20 proteins that were altered in the tumors and see whether many or most were altered at the RNA level. If they are this would greatly expand the potential use of genomic changes to examine for altered proteins which might be candidates for urine or other biomarkers. One obviously would not need a 1:1 correspondence between protein expression in tumors and expression of RNA .
The authors mention that the approach might be used in an initial screen and to discern invasive bladder cancer from TCC. However certainly in a blind screening would you put a premium on that discrimination or would you primarily be interested in differentiating tumor from normal. You may have more candidates to purely do that. Aso if looking for recurrence of a specific tumor would you be more interested in proteins that mght cover all tumors or rather a number of proteins which might be expressed at high levels in the tumor which might be indicative of a recurrence.?
Since the most common examination of expression is by RNA. It might be useful to determine whether some of the proteins which you observed altered in tumors may correspond to subgroups determined by genomics. Thus it is clear that roughly 25% of invasive bladder cancers fall into the basal category. the rest being luminal or luminal with altered P53 or other subgroups. Did certain of the proteins that you saw in the more advanced cancers correspond to the various subgroups. Thus it might be useful to know potential subgroups based on urine proteomics since that might affect therapy.
Author Response
Dear reviewer,
Please find below detailed answers to all of the raised points.
Sincerely,
Dr. Katarina Davalieva
Comments by the Reviewer 2
- The reviewer feels that there might be additional useful insights that might be gained by a comparison of the present data with the more commonly employed genomic approaches to bladder cancer. First using both their primary candidates NNMT, Galk-1 and HTRA are the differences observed in the urine and tissue at the protein level reflected in the RNA expression level. One might wish to examine 10 or 20 proteins that were altered in the tumors and see whether many or most were altered at the RNA level. If they are this would greatly expand the potential use of genomic changes to examine for altered proteins which might be candidates for urine or other biomarkers. One obviously would not need a 1:1 correspondence between protein expression in tumors and expression of RNA.
Thank you on this comment. The testing of the transcription levels of NNMT, GALK1, HTRA1 and other candidates tightly correlated with BCa stage is certainly the next step. We agree that this could led to the development of more speedy and cost effective test. However, this was not the aim of this study which is a proof-of concept study aimed to investigate alteration on protein level.
- The authors mention that the approach might be used in an initial screen and to discern invasive bladder cancer from TCC. However certainly in a blind screening would you put a premium on that discrimination or would you primarily be interested in differentiating tumor from normal. You may have more candidates to purely do that. Also if looking for recurrence of a specific tumor would you be more interested in proteins that might cover all tumors or rather a number of proteins which might be expressed at high levels in the tumor which might be indicative of a recurrence?
You are right, the initial aim was identification of proteins associated with BCa progression. Based on this we have designed the comparative proteomics study. However, testing the NNMT, GALK1 and HTRA1 in urine and including normal controls to BCa subgroups revealed that NNMT and GALK showed significant differences in protein levels between Controls and BCa with AUC of 1.000 (95% CI: 1.000-1.000; p < 0.0001) for NNMT and 0.801 (95% CI: 0.655-0.947; p < 0.0001) for GALK1 (Please see Figure 6A). In this context, NNMT was superior as it showed that can discriminate controls from even Ta G1 tumors, in addition to discriminating between BCa subgroups (Figure 5). However, future testing of the remaining 12 biomarker candidates significantly correlated with cancer stage with steady up or down with disease stages, could reveal even more candidates for BCa detection at early stages. We consider this study as a platform that enables more research into BCa biomarkers. We could not obviously test all of the candidates and there is no need for that at this stage as we have already proved the validity of our approach on the 3 candidates that we chose for validation.
- It might be useful to determine whether some of the proteins which you observed altered in tumors may correspond to subgroups determined by genomics. Thus it is clear that roughly 25% of invasive bladder cancers fall into the basal category, the rest being luminal or luminal with altered P53 or other subgroups. Did certain of the proteins that you saw in the more advanced cancers correspond to the various subgroups?
The genomic characterization of the somatic mutations present in the samples from our biobank have just started. We have currently profiled by NGS only 34 samples. Therefore, at present we don’t have the complete genomic data of the samples that were profiled by proteomics and therefore cannot do the correlations.
- It might be useful to know potential subgroups based on urine proteomics since that might affect therapy.
We agree with you, however, in this study we haven’t performed comparative proteomics profiling on urine samples, but just targeted measurement of 3 candidates. Therefore, further subgrouping is not possible based on the data we have. However, we plan to explore further the findings from this preliminary study, in particularly to validate NNMT, GALK1, HTRA1 and other candidates tightly correlated with BCa stage in bigger cohort of urine samples, which could possibly reveal subgroups associated with different outcomes and therapy.
Round 2
Reviewer 1 Report
The authors addressed all the concern raised by the reviewer and therefore the manuscript can be published.
Moderate editing of English language required